# Effect of Reinforcement Ratio and Bond Characteristic on Flexural Behavior of Carbon Textile-Reinforced Concrete Panels

**DOI:** 10.3390/ma16103703

**Published:** 2023-05-12

**Authors:** Jun-Mo Yang, Jongeok Lee, Chunho Chang

**Affiliations:** Department of Civil Engineering, Keimyung University, 1095 Dalgubeol-daero, Dalseo-gu, Daegu 42601, Republic of Korea; jm.yang@kmu.ac.kr (J.-M.Y.); eok129@kmu.kr (J.L.)

**Keywords:** textile-reinforced concrete, carbon fabric, flexural behavior, reinforcement ratio, bond

## Abstract

Textile-reinforced concrete (TRC) is highly anticipated as an alternative to reinforced concrete due to its ability to enable lightweight design, free formability, and improved ductility. In this study, TRC panel test specimens were fabricated and four-point loading flexural tests were performed to examine the flexural characteristics of TRC panels reinforced with carbon fabric, and to investigate the effect of the fabric reinforcement ratio, anchorage length, and surface treatment of fabric. Furthermore, the flexural behavior of the test specimens was numerically analyzed using the general section analysis concept of reinforced concrete and compared with the experimental results. Due to bond failure between the carbon fabric and the concrete matrix, the TRC panel showed a large decrease in flexural performance in terms of flexural stiffness, flexural strength, cracking behavior, and deflection. This low performance was improved by increasing the fabric reinforcement ratio, anchoring length, and sand–epoxy surface treatment of the anchorage. Comparing the numerical calculation results with the experimental results, the deflection of the experimental results was approximately 50% larger than the numerical calculation results. This is because the perfect bond between the carbon fabric and the concrete matrix failed, and slip occurred.

## 1. Introduction

Reinforced concrete is a globally used construction material that consists of homogeneous reinforcing bars and heterogeneous concrete. It is a composite material that offers various advantages in terms of material and structure by complementing different mechanical properties through the interaction of compression and tension. Based on these advantages, reinforced concrete plays a significant role in building modern society. With the recent developments of technology, there is a trend towards constructing larger and ultra-high reinforced concrete structures. However, this trend also leads to an increase in load due to the weight of the structure and external force, which can cause cracks on the surface of the reinforced concrete, leading to corrosion of the reinforcement and concrete. Corrosion of reinforcing bars and concrete can result in a deterioration of durability and stability, leading to significant repair and reinforcement costs. Therefore, there is a need for a replacement material for reinforced concrete to prevent a reduction in the durability and stability of structures, and to promote sustainable development in the construction industry; further, the interest in alternative materials is also increasing [1,2].

Textile-reinforced concrete is being explored as an alternative material for reinforced concrete. It is a composite material composed of a high-performance cement-based matrix, usually with a small aggregate maximum size, and a bundle of high-toughness continuous multifilaments composed of glass, carbon, polymer, or other materials [3]. Textile-reinforced concrete has a relatively small cover thickness compared to reinforcing bars, enabling lightweight design and free molding, which offers significant advantages. These mechanical properties allow fabric-reinforced concrete to be applied to reduce the weight of newly constructed structures and to repair/reinforce old structures [4,5,6]. In addition, textile-reinforced concrete can be applied in the field of 3D concrete printing. The textile reinforcement can be embedded at the same time as the concrete layers are 3D printed, so the textile reinforcements in each layer overlap in the interlayer direction to simulate continuous reinforcement [7].

Textile-reinforced concrete has numerous advantages and efficient application targets, but clear design criteria and guidelines have not yet been established, resulting in experiments being conducted based on the experience of researchers. As it is a composite material composed of two different materials, there are difficulties in understanding the material properties compared to conventional construction materials. Furthermore, because the adhesion behavior between the two materials, textile and concrete, has not been clearly identified, there are many limitations in analyzing the composite behavior and applying it to actual structures [8,9].

According to the literature, the main failure of TRC is caused by the slippage of the textile and the separation of the matrix and the textile, and this failure mode is related to the incomplete bonding of the textile and the matrix [10,11,12]. A study by de Felice et al. [13] revealed that the adhesion performance was not directly related to the textile tensile strength or stiffness but depended on the interaction between the textile and the matrix. Santis et al. [14] and Bertolesi et al. [15] conducted direct tensile tests on TRC coupons, and Sneed et al. [16] and Askouni et al. [17] conducted single/double-lap shear tests to investigate the local bond characteristics between textile and matrix. However, these studies can only provide macroscopic behaviors such as failure modes, force–slip curves, maximum elongation, and peak loads, and contain many unresolved issues.

In this study, to examine the difference in flexural behavior according to the textile reinforcement ratio, anchorage length, and surface treatment of concrete composite panels reinforced with carbon fabric, textile-reinforced concrete was fabricated and a flexural test performed. Differences in ultimate load, center deflection, and crack distribution ac-cording to each variable were compared and evaluated. In addition, the bending behavior was numerically calculated according to the nominal flexural strength calculation method of general reinforced concrete members and the deflection calculation method using the effective second moment of inertia, and the results were compared with the experimental results.

## 2. Experimental Activity

### 2.1. Details of Test Specimens

The experiments were divided into two parts: Group I and Group II. A total of 15 variables were set, with 5 variables in the Group I experiment and 10 variables in the Group II experiment. Three specimens were produced for each variable, resulting in a total of 45 TRC panel specimens that were manufactured and tested. Details of the TRC panel specimens are presented in Table 1.

There were three parameters: the number of carbon-fiber textile layers, the anchorage length of the textile, and the presence or absence of sand–epoxy surface treatment on the textile. The number of carbon-fiber textile layer parameters was set to 0 (NF), 1 (C1)∼4 (C4) to examine the effect of the textile reinforcement ratio on the flexural behavior of the TRC panel. Section details according to the number of textile layers are shown in Figure 1. A total of 5 variables were set in the range of 50 mm (A0) to 275 mm (A4) for the anchorage length parameter. Since the bond failure phenomenon, in which the fabric and the concrete matrix are separated, is prominently observed in the carbon-fiber textile [9], the effect of anchorage length on the flexural behavior of the TRC panel was investigated. In addition, as a method for secure anchoring of the textile, specimens (SE) treated with sand–epoxy in the anchorage length section were set as the third parameter and tested. The warp was placed in the longitudinal direction of the test specimen so that the warp of the textile was subjected to longitudinal stress in all specimens.

The width of the TRC panel specimens was fixed at 100 mm, and the height (*h*) was based on 40 mm, but its height slightly varied depending on the number of textile layers. The actual measured height of the specimen and the location of each textile layer are presented in detail in Table 1. The span length of the four-point bending test was fixed at 300 mm, and the anchorage lengths on both sides were changed from 50 mm to 275 mm depending on the anchorage length variable. Therefore, the TRC panel test specimens were differently manufactured with a total length of 400 mm to 850 mm (Figure 2). The TRC panel test specimens were manufactured by the casting method, and the manufacturing procedure is shown in Figure 3. First, a mold was made from plywood (Figure 3a); then, a release agent was applied (Figure 3b). The carbon textile was cut according to the mold size, and a layer of fabric and the upper mold were alternately arranged and fixed (Figure 3c). The concrete matrix was mixed and poured (Figure 3d), cured for more than 7 days after completion of the pouring (Figure 3e), and the mold removed (Figure 3f).

### 2.2. Materials

Textile-reinforced concrete is typically produced by laminating two-dimensional textiles in multiple layers with concrete. The two-dimensional textile is woven by crossing the warp and weft yarns, with the load mainly applied to the warp and the weft playing a role in maintaining the gap between the warp yarns. For this study, a carbon-fiber textile was used, as shown in Figure 4. The warp yarn with 10 mm center spacing was woven with 48K carbon fiber, and the weft yarn with 15 mm spacing was woven with 12 K carbon fiber. The surface was coated with styrene butadiene. The mechanical properties of the carbon-fiber textile provided by the manufacturer are shown in Table 2. To increase the mechanical bonding between the textile and the concrete matrix, sand–epoxy treatment was applied to the anchorage length area of some textiles. Epoxy was applied first; then, silica sand with a size of 0.17–0.25 mm was sprayed. The surface condition of the textile before and after sand–epoxy treatment can be seen in Figure 4.

A commercially available non-shrinking hydraulic cement pre-mixed mortar, named Grout EM and produced by Union, was used to produce the concrete matrix of the TRC panel. Table 3 presents the test results of the mortar provided by the pre-mixed product manufacturer. A matrix was prepared by mixing the mortar powder and water in a weight ratio of 100:15.5 to create the TRC panel specimens. To measure the flexural and compressive strength of the matrix following the ISO 679 [18] and KS F 4044 [19] methods, 40×40×160 mm specimens were prepared. On the day of the flexural test of the TRC panel, the flexural strength of the matrix was measured by applying a central point load to three specimens of the matrix with a span length of 100 mm. After that, the compressive strength was measured using six fragments generated after the three flexural strength tests. Additionally, cylindrical specimens with a diameter of 100 mm and a length of 200 mm were manufactured to measure the compressive strength by the ASTM C39 [20] and KS F 2405 [21] methods. In the case of Group I, the stress–strain curve was obtained by attaching strain gauges to both sides of the cylindrical specimen, and the modulus of elasticity of the concrete matrix was calculated using the ASTM C469 [22] and KS F 2438 [23] methods. The measured mechanical properties of the concrete matrix for Group I and Group II are listed in Table 4.

### 2.3. Test Setup

The four-point bending test of the TRC panel was performed according to the setup illustrated in Figure 2, with a 300 mm distance between the lower supports and 100 mm for the upper loading points. The load was applied at a constant speed of 0.2 mm/min using a 500 kN UTM from MTS, Eden Prairie, MN, USA at Intelligent Construction System Core-Support Center, Keimyung University, Republic of Korea, and the vertical displacement was measured by installing LVDT on both sides of the mid-span while recording the load value in the UTM. If the load decreased to 70% or less of the maximum load, the test was terminated.

## 3. Numerical Calculation of TRC Panel

### 3.1. Flexural Strength

In order to analyze the flexural strength of the carbon-fiber textile-reinforced concrete panel, the flexural strength calculation method for the general reinforced concrete beam was used, and the numerically calculated values for each specimen were compared with the results of the flexural test. The numerical calculation was based on the following basic assumptions in flexure theory [24,25].

(1)Sections perpendicular to the axis of bending that are plane before bending remain plane after bending.(2)The concrete matrix and textile are fully bonded, so the strain in the textile is equal to the strain in the concrete matrix at the same level.(3)The stress–strain relationship of carbon-fiber textile is perfectly linear.(4)The tensile strength of concrete and compressive strength of carbon-fiber textile are neglected in flexural strength calculations.(5)Stress induced by flexural load acts only on the warp of the textile reinforcement.

For the numerical calculation, the Hognestad [26] model was used as a compression model for concrete, and the stress–strain relationship in concrete was assumed to be a quadratic parabola, as shown in Equation (1).
(1)$$\sigma_{c}=f_{c}^{\prime }\left[2\frac{\varepsilon_{c}}{\varepsilon_{c}^{\prime }}-{\left(\frac{\varepsilon_{c}}{\varepsilon_{c}^{\prime }}\right)}^{2}\right]$$
where $\varepsilon_{c}$ = the compressive strain of the matrix, $\sigma_{c}$ = compressive stress corresponding to $\varepsilon_{c}$, $f_{c}^{\prime }$ = the compressive strength of the matrix, $\varepsilon_{c}^{\prime }$ = the compressive strain of the matrix corresponding to $f_{c}^{\prime }$, and $\varepsilon_{u}$ = the ultimate compressive strain of the matrix. The $\varepsilon_{c}^{\prime }$ and $\varepsilon_{u}$ were assumed to be 0.0028 and 0.0038, respectively, based on the stress–strain relationship curve derived from the elastic modulus test for the Group I matrix.

According to the assumption that the plane section remains plane, the strain of each textile layer and the compressive strain of concrete are proportional to the distance away from the neutral axis, as shown in Figure 5, and are determined according to Equations (2) and (3).
(2)$$\varepsilon_{f1}=\frac{h_{f1}-h_{c}}{h-h_{c}}\varepsilon_{tm}\;;\;\varepsilon_{f2}=\frac{h_{f2}-h_{c}}{h-h_{c}}\varepsilon_{tm};{\;\varepsilon }_{f3}=\frac{h_{f3}-h_{c}}{h-h_{c}}\varepsilon_{tm};{\;\varepsilon }_{f4}=\frac{h_{f4}-h_{c}}{h-h_{c}}\varepsilon_{tm}$$
(3)$$\varepsilon_{cm}=-\frac{h_{c}}{h-h_{c}}\varepsilon_{tm}$$
where $h_{f1}$, $h_{f2}$, $h_{f3}$, $h_{f4}$ = the distance from the edge of the compression zone to the first, second, third, and fourth layer of the textile, respectively; $\varepsilon_{f1}$, $\varepsilon_{f2}$, $\varepsilon_{f3}$, $\varepsilon_{f4}$ = the strain of the first, second, third, and fourth layer of the textile, respectively; h = the height of the cross section of the TRC panel; $h_{c}$ = the height of the compression zone (distance from the compression edge to the neutral axis); $\varepsilon_{tm}$ = the tensile strain of the matrix at the edge of the tension zone t; and $\varepsilon_{cm}$ = the compressive strain of the matrix at the edge of the compression zone.

Equation (4) can be derived using the force equilibrium condition, and $h_{c}$ can be obtained from Equation (4). In addition, Equation (5) can be derived using the equilibrium condition of the moment with respect to the neutral axis, and the resistant moment, M, can be obtained.
(4)$$\int_{0}^{h_{c}}b\sigma_{c}\left(x\right)dx+E_{f}A_{f}\left(\varepsilon_{f1}+\varepsilon_{f2}+\varepsilon_{f3}+\varepsilon_{f4}\right)=0$$
(5)$$M=\int_{0}^{h_{c}}b\sigma_{c}\left(x\right)xdx-E_{f}A_{f}\left\{\varepsilon_{f1}\left(h_{f1}-h_{c}\right)+\varepsilon_{f2}\left(h_{f2}-h_{c}\right)+\varepsilon_{f3}\left(h_{f3}-h_{c}\right)+\varepsilon_{f4}\left(h_{f4}-h_{c}\right)\right\}$$
where $\sigma_{c}\left(x\right)$ = the compressive stress function of the matrix.

When the compressive strain of the matrix reaches the ultimate compressive strain, the TRC panel fails and the resistance moment at that time corresponds to the nominal moment, $M_{n}$, of the panel.

### 3.2. Midspan Deflection

The deflection of the carbon-fiber textile-reinforced concrete panel was calculated using the elastic deflection equation. As for the moment of inertia applied to the elastic deflection equation, the effective moment of inertia, $I_{e}$, of Equation (6)—proposed by ACI Committee 440 [27] as a method for predicting the deflection of the FRP bar-reinforced concrete beams—was applied. Then, it was compared with the test results of the carbon-fiber textile-reinforced concrete panel.
(6)$$I_{e}=\frac{I_{cr}}{1-\gamma {\left(\frac{M_{cr}}{M_{a}}\right)}^{2}\left[1-\frac{I_{cr}}{I_{g}}\right]}\;\leq \;I_{g}$$
where $I_{g}\;$= the gross (uncracked) moment of inertia, $I_{cr}$ = the cracked transformed moment of inertia, $M_{cr}$ = the cracking moment, and $M_{a}=$ the service load moment at the critical section.

In order to predict the deflection of reinforced concrete flexural members, ACI Committee 318 [24] uses the concept of the effective moment of inertia proposed by Branson [28]. However, Bischoff and Scanlon [29], and Bischoff [30], pointed out that the Branson equation underestimates the deflection of not only FRP-reinforced concrete beams, but also steel-reinforced concrete beams with a reinforcement ratio of less than 1%; then, they proposed a new equation of effective moment of inertia. In addition, an additional factor γ is added to the equation, and Equation (6) is obtained. The factor γ is to account for the variation in stiffness along the length of the member and can be obtained by the integration of curvature along the span [30,31]. In the case of the third point load and simply supported beam condition, which correspond to the flexural test condition of this paper, the factor γ can be taken as $1.7-0.7\left(M_{cr}/M_{a}\right)$ [31].

## 4. Test Result and Discussion

### 4.1. Flexural Behavior

The load–deflection curves were illustrated for the three TRC panel specimens for each variable resulting from the flexural test. The load–deflection relationships for the three specimens were averaged and are presented as one graph for each variable. For instance, in the case of the A4SE series test specimens, Figure 6a shows the experimental results for four variables and 12 specimens, while the average load–deflection curve for each variable is shown in Figure 6b. Additionally, the results of the flexural test for each variable were averaged and are presented in Table 3.

The NF-I and NF-II test specimens, which did not have fabric reinforcement, showed a linear increase in vertical deflection as the load increased and failed as soon as a flexural crack occurred between the two-point loading parts, with an average ultimate load of 3.68 kN and 3.01 kN, respectively. The difference in flexural strength between the NF-I and NF-II test specimens appears to be due to the difference in the compressive strength of the concrete matrix (Table 4).

The carbon fabric-reinforced TRC panel specimens exhibited the same initial behavior as the NF specimens, then developed flexural cracks and continued to behave without failure. The first flexural cracking load did not show any variation across the variables. All of the TRC specimens showed a sudden drop in load immediately after the first flexural crack occurred, a phenomenon resulting from the brittle stress redistribution from the concrete matrix to the carbon fabric [8]. This phenomenon was observed not only when the first crack occurred, but also when additional cracks, such as the second and third cracks, occurred.

### 4.2. Effect of the Number of Textile Layers

The average load–deflection relationship between the A0 series, which had the shortest anchorage length, and the A4SE series treated with sand–epoxy, is shown in Figure 7 and Figure 6b, respectively. In all specimens, a decrease in load was observed when flexural cracks occurred. Compared to specimens reinforced with one layer of carbon fabric, the extent of load reduction was reduced in the case of specimens reinforced with two, three, and four layers. This reduction can be recovered by increasing the reinforcement ratio of the fabric or by increasing the bond strength between the fabric and concrete, so that the amount of fabric that can receive stress is increased and the stress can be transmitted smoothly [1]. The TRC panel specimens showed a tendency for the flexural stiffness to decrease compared to the flexural stiffness before cracking, as the reduced load after the first crack was recovered. When additional flexural cracks occurred, the flexural stiffness further decreased. The reduction rate of flexural stiffness was greater when the reinforcement ratio of the fabric was lower. In a state where the flexural stiffness was very low, there was a phenomenon in which only the displacement increased greatly with little change in load. This seems to be because the carbon fabric separates from the concrete matrix when the tensile stress applied to the fabric exceeds the bond strength, causing both ends of the fabric to be gradually pulled towards the center of the specimen. This phenomenon was more pronounced when the reinforcement ratio of the fabric was lower.

The maximum load was the highest for the C4L1A0 test specimen in the A0 series, with an average of 13.51 kN, and for the C4L1A4SE specimen in the A4SE series, with 27.45 kN. Both the A0 series and the A4SE series demonstrated higher maximum loads as the number of carbon fabric reinforcing layers increased (Table 3). The C4L1A0 and C4L1A4SE test specimens, which had four layers of carbon fabric, exhibited a 227% and 216% increase in maximum load compared to the C1L1A0 and C1L1A4SE specimens with the lowest reinforcement ratio. This is because the C4L1A0 and C4L1A4SE test specimens had the fabric placed 17–18 mm lower than the C1L1A0 and C1L1A4SE specimens, resulting in a rapid contribution to the tensile stress acting on the lower part of the concrete matrix.

The crack patterns observed after completing the flexural test on the TRC panel specimens are presented in Figure 8. The A4SE series showed a clear tendency of crack pattern according to the fabric reinforcement ratio. In all specimens, more than two flexural cracks occurred, with an average number of cracks of 2.3, 3, 4, and 4 for the C1L1A4SE, C2L1A4SE, C3L1A4SE, and C4L1A4SE specimens, respectively. The highest number of cracks was observed in the C3L1A4SE and C4L1A4SE specimens. This is because the fabric was placed closer to the bottom fiber compared to the C1L1A4SE and C2L1A4SE specimens, resulting in smoother stress transfer from the concrete to the fabric. Additionally, as the fabric reinforcement ratio to tensile stress increases, the number of flexural cracks generally increases [2]. Therefore, it is possible that placing the fabric closer to the bottom of the specimen and increasing the fabric reinforcement ratio can increase the flexural stiffness and maximum load, leading to an increase in the number of flexural cracks and, ultimately, an increase in the flexural toughness.

### 4.3. Effect of the Textile Anchorage Length

The load–deflection curve according to the difference in anchorage length is shown in Figure 9. It was found that the flexural stiffness after cracking increased as the anchorage length increased. The maximum load also tended to increase as the anchorage length increased, but the C2L1A4 specimen with an anchorage length of 275 mm showed a lower maximum load than the C2L1A3 specimen with an anchorage length of 225 mm. This indicates that there is a limit to the increase in anchorage length required to improve the adhesion behavior between concrete and fabric. The maximum load of the C2L1A3 specimen increased by 52% compared to that of the C2L1A0 specimen.

The number of cracks in the C2L1A0, C2L1A1, C2L1A2, C2L1A3, and C2L1A4 specimens were 2.3, 2.3, 2.7, 3.7, and 3.7, respectively, indicating an increase in the number of cracks with longer anchorage lengths. For the C2L1A0 specimen, large crack widths were observed, and carbon fabric slip between the cracks was visible. However, as anchorage length increased, crack widths decreased.

### 4.4. Effect of the Sand–Epoxy Surface Treatment on Textile

Figure 10 shows the load–deflection relationship with and without sand–epoxy surface treatment. The C1L1A4SE, C2L1A4SE, C3L1A4SE, and C4L1A4SE test specimens showed a maximum load increase of 110%, 60%, 79%, and 106%, respectively, compared to the C1L1A0, C2L1A0, C3L1A0, and C4L1A0 specimens. This increase was due to the simultaneous effect of an increase in anchorage length and the sand–epoxy surface treatment. The C2L1A1 test specimen had a small increase in anchorage length of 75 mm compared to the C2L1A0 specimen, resulting in a 19% increase in maximum load. However, through the sand–epoxy surface treatment, the maximum load increased by an additional 79%, making the effect of the sand–epoxy surface treatment more significant than the increase in anchorage length (Figure 10b). On the other hand, the C2L1A4 test specimen had a significant increase in anchorage length of 225 mm compared to the C2L1A0 specimen, resulting in a 43% increase in maximum load. The additional increase in maximum load through sand–epoxy surface treatment was 17%, making the effect of the anchorage length greater than that of the sand–epoxy surface treatment (Figure 10c). The C2L1A1 specimen without sand–epoxy surface treatment exhibited a 21% increase in maximum load due to the increase in anchorage length compared to the C2L1A4 specimen. However, the C2L1A1SE specimen with sand–epoxy surface treatment, which also had an increased anchorage length of C2L1A4SE, had a 19% reduction in maximum load (Figure 10d). This may be because the effectiveness of improving load-carrying capacity due to the increase in anchorage length was mostly offset by the sand–epoxy surface treatment.

The C1L1A4SE and C2L1A4SE specimens experienced flexural failure, while the C3L1A4SE and C4L1A4SE specimens developed flexural shear cracks that subsequently led to flexural shear failure. Flexural cracks occur due to tensile stress that arises in the vertical direction at the bottom of the specimen. The reduction in cross-sectional area resulting from the occurrence of flexural cracks causes an increase in inclined stress and shear stress at the tip of the flexural crack, which expands and leads to a flexural shear crack. The C1L1A4SE and C2L1A4SE specimens had a low carbon fabric reinforcement ratio, resulting in long flexural cracks towards the compression fiber, which made it difficult for them to expand into inclined cracks. In contrast, the C3L1A4SE and C4L1A4SE specimens could easily expand into inclined cracks due to the controlled expansion of flexural cracks, as the carbon fabric received the tensile stress adequately due to the increased carbon fabric reinforcement ratio. Upon examining the crack patterns of the SE series, the number of cracks increased, while the crack width decreased due to the sand–epoxy surface treatment.

## 5. Comparison of Experimental Results and Numerical Calculation Results

The load–deflection relationship curves of the TRC panel specimens were plotted using the two equations proposed by Branson and Bischoff. The flexural moment corresponding to the assumed $\varepsilon_{cm}$ was obtained by using Equation (5), and the elastic deflection was obtained using the effective moment of inertia at that $\varepsilon_{cm}$. $\varepsilon_{cm}$ was subdivided from 0 to 0.0038, and load and deflection for each $\varepsilon_{cm}$ were all obtained and connected with a line to derive a load–deflection relationship curve. These were then compared with the results of the flexural test of four specimens with sand–epoxy surface treatment, as shown in Figure 11. Additionally, the representative values derived through numerical calculation were compared with the experimental values and presented in Table 5.

The initial behavior of the numerical calculation and the experimental results was very similar in all specimens. The behavior before the first flexural crack occurred showed identical results, but the time of occurrence of the first flexural crack was slightly different. This was because the load at which the first flexural crack occurred in the TRC panel test specimen was mostly higher than the flexural strength of the concrete matrix itself (Table 5).

The behavior of all test specimens after the first flexural crack was closer to the experimental results when the Bischoff formula was applied than when the Branson formula was used. As the carbon fabric reinforcement ratio increased, the difference between the application results of the Branson formula and the Bischoff formula decreased [29]. The behavior after the first flexural crack showed that the flexural stiffness of the experimental results was about 50% smaller than the numerical calculation. This seems to be because the basic assumption mentioned in Clause 3.1, “The concrete matrix and textile are fully bonded, so the strain in the textile is equal to the strain in the concrete matrix at the same level”, was applied in the numerical calculation, whereas the perfect bond between the concrete matrix and the carbon fabric was not achieved in the experiment. In the test section between the lower supports, the bond strength between the carbon fabric and the concrete matrix was lowered, resulting in non-integrated behavior, and slip occurred, leading to increased deflection. Such bond failure results are particularly evident in coated fabrics and fabrics composed of carbon fibers [9]. Comparing deflection at a load of 33% of the nominal strength, the deflection resulting from the experiment was 40% to 59% larger than the numerically calculated deflection (Table 5). In order to prevent such slippage and bond failure, methods such as surface improvement through sand–epoxy surface treatment, sufficient anchorage length, and concrete matrix thickness are required. To design structures applying TRC, it is necessary to accurately identify the bond behavior and set limitations for TRC material performance. Alternatively, TRC behavior can be predicted by assuming incomplete adhesion between the carbon fabric and the concrete matrix as a reduced modulus of elasticity of the carbon fabric [32].

In the case of maximum load, the experimental results were 8% to 22% higher when compared to the nominal strength by numerical calculation, except for the C2L1A4SE specimen. This difference can be attributed to the variation in ultimate compressive strain assumed in the numerical calculation and the difference in the first cracking load.

## 6. Conclusions

To investigate the effect of the fabric reinforcement ratio, anchorage length, and surface treatment on concrete composite panels reinforced with carbon fabric, textile-reinforced concrete specimens were fabricated and flexural tests conducted. In addition, the flexural behavior of the TRC panel specimens was numerically calculated and compared with the experimental results, and the following conclusions were drawn.

(1)All TRC specimens exhibited a sudden drop in load immediately after the first flexural crack occurred. This is due to the brittle stress redistribution from the concrete matrix to the carbon fabric, and it can be reduced by improving the bond between the matrix and the carbon fabric and increasing the fabric reinforcement ratio.(2)With an increase in the carbon fabric reinforcement ratio, from one layer to four layers, the degree of load reduction and reduction in flexural stiffness at the time of flexural cracks decreased, and the maximum load and number of cracks increased. This is because the carbon fabric rapidly contributes to the tensile stress acting on the bottom of the concrete matrix.(3)As the anchorage length increased from 50 mm to 275 mm, the flexural stiffness after the first cracking of the TRC panel increased, and the number of cracks increased.(4)Sand–epoxy surface treatment and an increase in anchorage length simultaneously contributed to the improvement of the load-carrying capacity of the TRC panel up to 110%. However, the effect of increasing the anchorage length was limited, and additional load-bearing capacity could be improved through sand–epoxy surface treatment.(5)Comparing the numerical calculation results with the experimental results, it was found that the deflection of the experimental results was approximately 50% larger than the numerical calculation results. This is because the bond between the carbon fabric and the concrete matrix failed, and slip occurred.

To prevent slip of the fabric, it is necessary to secure a reliable bond between the carbon fabric and concrete using methods such as surface improvement through sand–epoxy surface treatment, sufficient anchorage length, and sufficient concrete thickness. In addition, it is necessary to accurately identify the bond behavior and set limitations for TRC material performance to design structures applying TRC.

## Figures and Tables

**Figure 1 materials-16-03703-f001:**
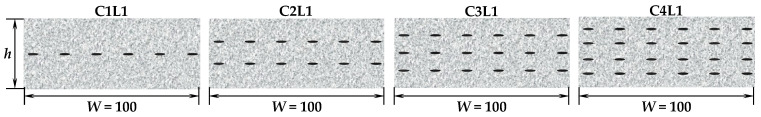
Section details of TRC panel specimens.

**Figure 2 materials-16-03703-f002:**
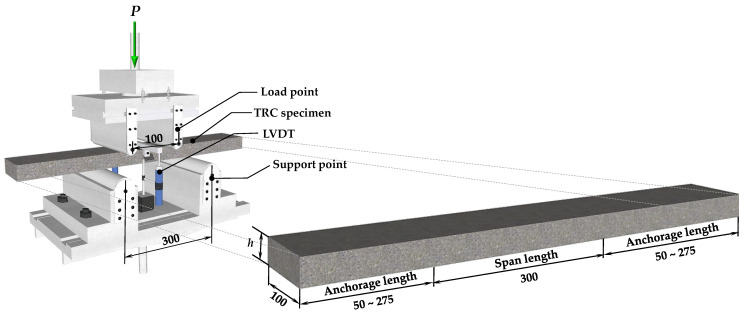
Details of TRC test specimen and four-point bending test setup (the unit of length is mm).

**Figure 3 materials-16-03703-f003:**
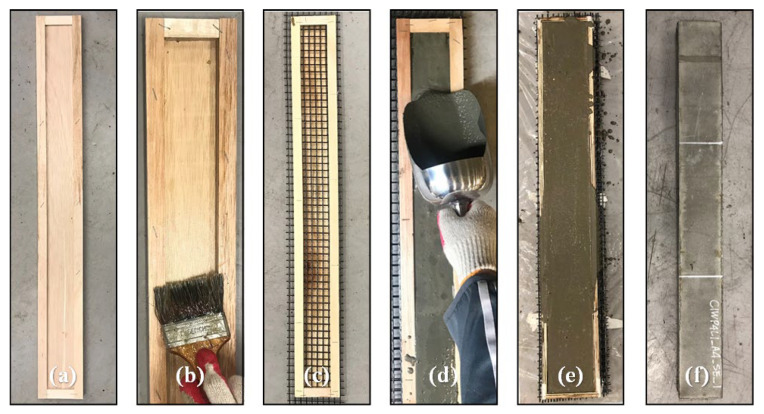
Manufacturing procedure of TRC specimen: (**a**) mold; (**b**) applying release agent; (**c**) arranging fabric; (**d**) concrete casting; (**e**) curing; (**f**) demolding.

**Figure 4 materials-16-03703-f004:**
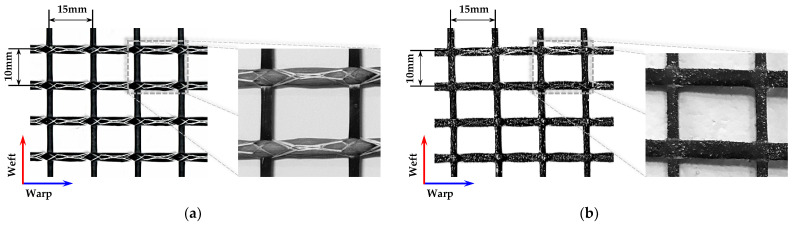
Details of carbon-fiber textile: (**a**) carbon-fiber textile without sand–epoxy treatment; (**b**) carbon-fiber textile with sand–epoxy treatment.

**Figure 5 materials-16-03703-f005:**
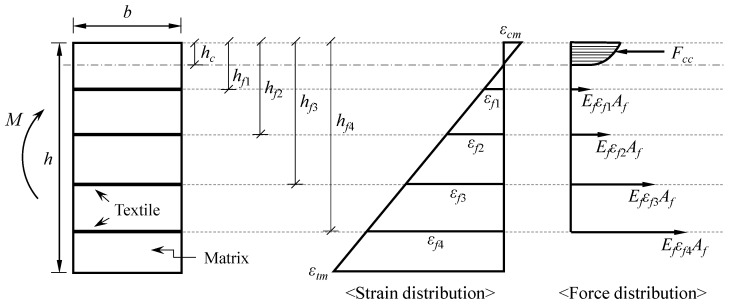
Strain and force distributions of TRC specimen.

**Figure 6 materials-16-03703-f006:**
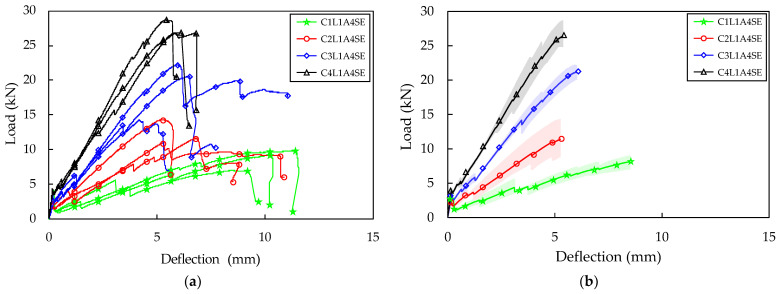
Four-point bending test result (A4SE series): (**a**) total load–deflection curve; (**b**) average load–deflection curve.

**Figure 7 materials-16-03703-f007:**
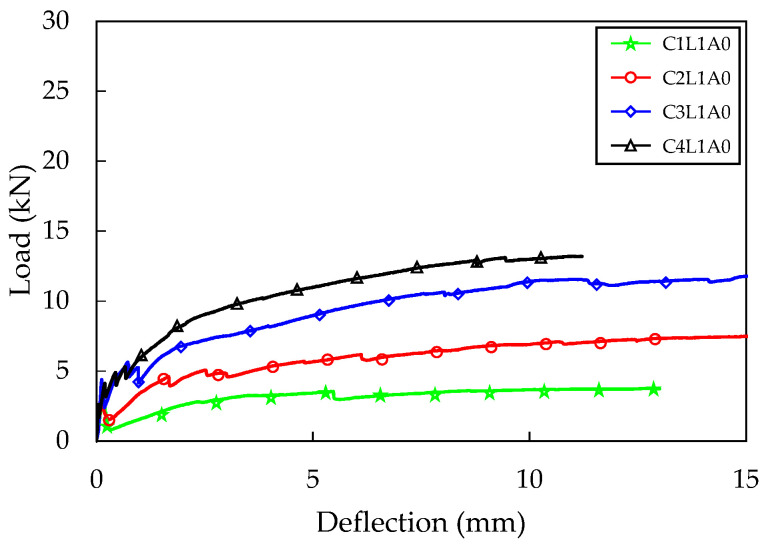
Four-point bending test result (A0 series).

**Figure 8 materials-16-03703-f008:**
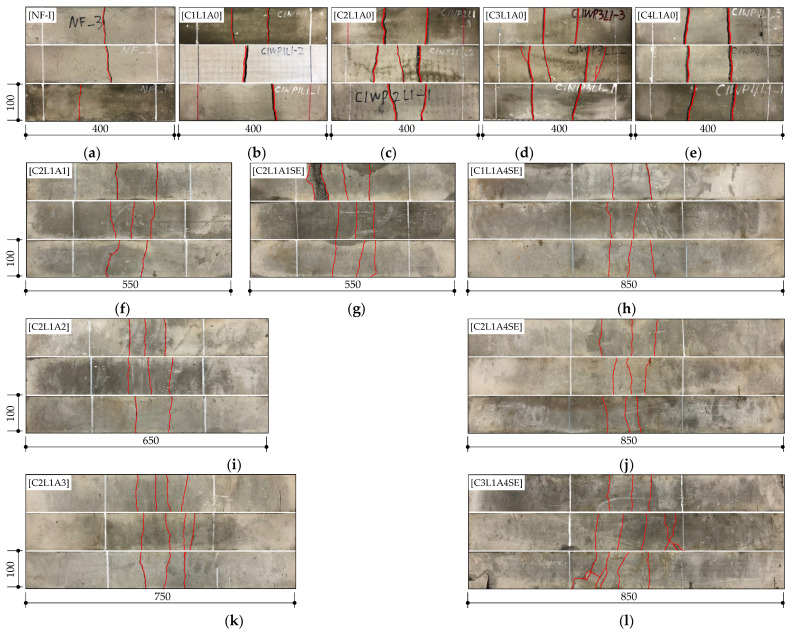
Crack patterns (the unit of length is mm): (**a**) NF-I; (**b**) C1L1A0; (**c**) C2L1A0; (**d**) C3L1A0; (**e**) C4L1A0; (**f**) C2L1A1; (**g**) C2L1A1SE; (**h**) C1L1A4SE; (**i**) C2L1A2; (**j**) C2L1A4SE; (**k**) C2L1A3; (**l**) C3L1A4SE; (**m**) C2L1A4; (**n**) C4L1A4SE.

**Figure 9 materials-16-03703-f009:**
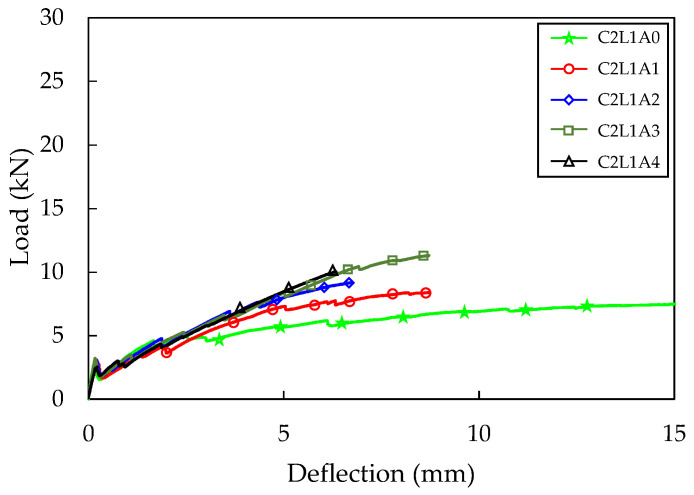
Four-point bending test result (C2L1 series).

**Figure 10 materials-16-03703-f010:**
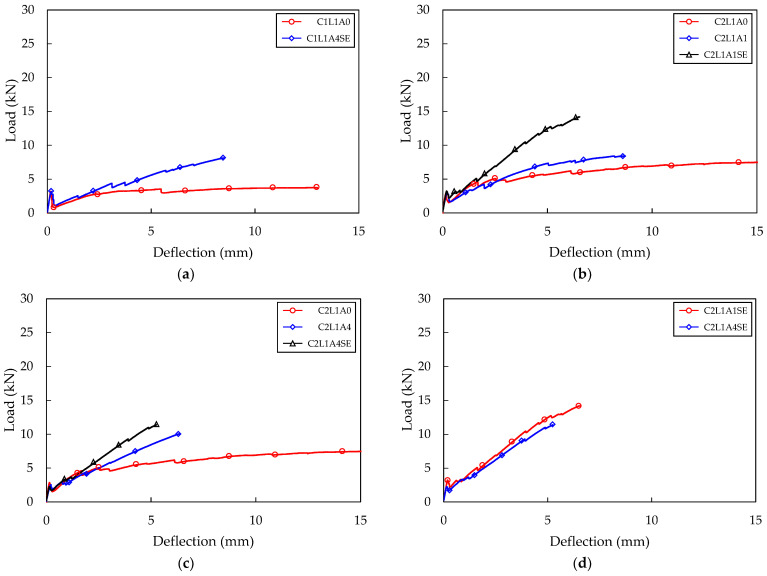
Four-point bending test result (SE series): (**a**) C1L1 series; (**b**) C2L1A1 series; (**c**) C2L1A4 series; (**d**) C2L1SE series; (**e**) C3L1 series; (**f**) C4L1 series.

**Figure 11 materials-16-03703-f011:**
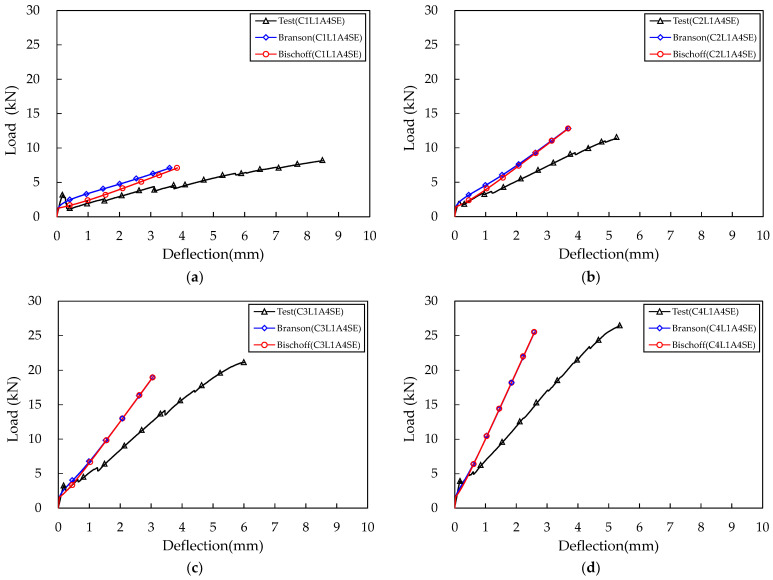
Comparison test and calculation result: (**a**) C1L1A4SE; (**b**) C2L1A4SE; (**c**) C3L1A4SE; (**d**) C4L1A4SE.

**Table 1 materials-16-03703-t001:** Details of test specimens.

Group	Specimen Name	Number of Fabric Layer	Anchorage Length (mm)	Sand–Epoxy Treatment	Height of Cross Section, *h* (mm)	Location of Each Fabric Layer from Top Concrete Fiber (mm)			
I	NF-I	0	-	-	40.0	-			
	C1L1A0	1	50	Not applied	41.6	20.9			
	C2L1A0	2	50	Not applied	44.0	12.4		27.3	
	C3L1A0	3	50	Not applied	45.6	9.7	21.8		32.7
	C4L1A0	4	50	Not applied	47.2	10.1	17.9	28.3	37.8
II	NF-II	0	-	-	40.0	-			
	C2L1A1	2	125	Not applied	44.2	11.9		27.1	
	C2L1A2	2	175	Not applied	44.6	12.3		27.4	
	C2L1A3	2	225	Not applied	44.4	12.0		27.1	
	C2L1A4	2	275	Not applied	44.6	12.1		27.2	
	C1L1A4SE	1	275	Applied	41.2	20.6			
	C2L1A4SE	2	275	Applied	44.1	11.9		27.2	
	C2L1A1SE	2	125	Applied	43.9	12.2		27.3	
	C3L1A4SE	3	275	Applied	46.2	9.5	21.5		33.3
	C4L1A4SE	4	275	Applied	47.6	9.3	18.7	28.5	38.5

**Table 2 materials-16-03703-t002:** Mechanical properties of carbon-fiber textile presented by manufacturer.

		Tensile Strength (MPa)	Elongation (%)	Section Area (mm^2^/m)	Density (g/m^3^)	Weight after Coating (g/m^2^)	Coating

Warp		2551	1.17	142	1.8	350	styrene butadiene
Weft		2847	1.24	25	1.8		

**Table 3 materials-16-03703-t003:** Properties of pre-mixed non-shrink mortar provided by manufacturer.

		Test Results	Test Method

Flow time by cone test		45 s.	KS F 4044 [19] Mix ratio (Pre-mixed binder: water) = 100 kg: 15.5 kg
Flow by spread test		Over 300 mm	
Setting time		Initial: 3 h., Final: 4 h. 15 min.	
Bleeding		0%	
Height of expansion		1 day: 0.02%, 3 day: 0.01%, 7 day: 0.00%, 28 day: 0.00%	
Compressive strength		1 day: 16.7 MPa, 3 day: 35.3 MPa, 7 day: 42.5 MPa, 28 day: 58.3 MPa	
Chloride contents		0.13 kg/m^3^	

**Table 4 materials-16-03703-t004:** Measured mechanical properties of the concrete matrix.

**Test Method**	**ISO 679 [18], KS F 4044 [19]**		**ASTM C39 [20],** **KS F 2405 [21]**	**ASTM C469 [22],** **KS F 2438 [23]**
	**Flexural Strength**	**Compressive Strength**	**Compressive Strength**	**Elastic Modulus**
	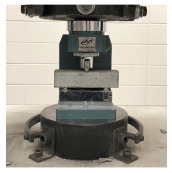	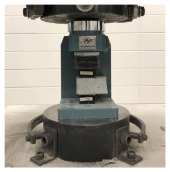	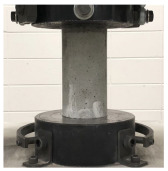	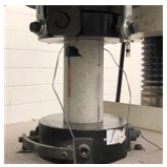
Group I	10.0 MPa	59.9 MPa	51.8 MPa	27.6 GPa
Group II	10.3 MPa	53.2 MPa	49.5 MPa	-

**Table 5 materials-16-03703-t005:** Four-point bending test result and comparison with calculation result.

Specimen Name	*I_g_*/*I_cr_*	Textile Reinforcing Ratio (%)	Experiment				Calculation		Ratio (1)/(3)	Ratio (2)/(4)
			First Cracking Load (kN)	Ultimate Load (kN) (1)	Deflection at 1/3 *M_n_* (mm) (2)	Toughness (N·m)	Ultimate Load (kN) (3)	Deflection at 1/3 *M_n_* (mm) (4)		
NF-I	-	-	3.68 (0.290)	3.68 (0.290)	-	-	1.65	-	2.23	-
C1L1A0	16.89	0.34	3.33 (0.009)	4.13 (0.097)	0.12 (0.131)	42.52 (0.135)	7.88	0.72	0.52	0.16
C2L1A0	12.10	0.64	3.39 (0.201)	7.71 (0.052)	1.69 (0.159)	113.28 (0.283)	13.98	1.14	0.55	1.48
C3L1A0	7.48	0.93	4.82 (0.015)	11.97 (0.075)	1.81 (0.023)	159.95 (0.041)	19.90	0.99	0.60	1.83
C4L1A0	5.13	1.20	4.52 (0.043)	13.30 (0.015)	2.33 (0.328)	127.90 (0.128)	26.29	0.84	0.51	2.77
NF-II	-	-	3.01 (0.086)	3.01 (0.086)	-	-	1.18	-	2.55	-
C2L1A1	12.14	0.64	3.25 (0.109)	9.16 (0.121)	2.03 (0.239)	209.53 (0.511)	12.79	1.11	0.72	1.83
C2L1A2	12.12	0.64	3.20 (0.257)	9.97 (0.015)	1.67 (0.291)	172.53 (0.527)	13.01	1.10	0.77	1.52
C2L1A3	12.29	0.64	3.25 (0.157)	11.70 (0.271)	2.42 (0.259)	149.42 (0.289)	12.78	1.10	0.92	2.20
C2L1A4	12.34	0.64	3.20 (0.106)	11.04 (0.118)	1.81 (0.215)	99.43 (0.274)	12.86	1.10	0.86	1.65
C1L1A4SE	18.78	0.34	3.37 (0.233)	8.69 (0.170)	1.11 (0.091)	102.63 (0.274)	7.10	0.97	1.22	0.11
C2L1A4SE	11.81	0.65	2.34 (0.174)	12.34 (0.142)	1.66 (0.237)	112.78 (0.266)	12.87	1.11	0.96	1.50
C2L1A1SE	11.67	0.65	3.46 (0.099)	15.28 (0.083)	1.27 (0.154)	133.01 (0.206)	12.94	1.11	1.18	1.15
C3L1A4SE	7.65	0.92	3.25 (0.168)	21.42 (0.053)	1.36 (0.162)	230.09 (0.412)	19.03	0.97	1.13	1.40
C4L1A4SE	5.28	1.19	4.50 (0.121)	27.45 (0.040)	1.33 (0.031)	189.91 (0.088)	25.53	0.84	1.08	1.59

( ): coefficient of variation for three specimens.

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
