# Peer review of "Effect of Reinforcement Ratio and Bond Characteristic on Flexural Behavior of Carbon Textile-Reinforced Concrete Panels"

_materials, 2023, doi:10.3390/ma16103703_

Round 1

Reviewer 1 Report

This manuscript is related to the effect of reinforcement ratio and bond characteristic on flexural behavior of carbon textile reinforced concrete panels. Some corrections are needed before publication.

The introduction is very general and recent studies should be addressed and details of flexural strength provided for other concretes should be reviewed and compared. The authors can refer to the recent studies: “Experimental investigation on the bond performance of sea sand coral concrete with FRP bar reinforcement for marine environments “; “Assessment of Diagonal Macrocrack-Induced Debonding Mechanisms in FRP-Strengthened RC Beams”;

What is the cement compositions?

Figure 6b add the error bar?

Complete the caption of the figure 8 and also add scale bar for each figure.

The fracture cross-section of samples can be very helpful in determining the failure mechanism. It is necessary to provide a microscopic image of the fracture cross-section of the samples and discussed related to the failure mechanism.

The references are very old, while many studies have been done in recent years and should be updated.

Table 4: Why the authors do not complete mechanical test with split tensile tests?

The materials and methods part is incomplete and they should provide complete information about concrete composition, particle size, cement-water ratio, setting time, etc.

Figure 10. Considering the noise in the graphs, it is recommended to use the regression method to redesign the graphs.

The conclusion should express the overall achievement of the article and be completed with numerical results.

Author Response

Please find attached file of the response to reviewer's comments.

Reviewer 2 Report

The manuscript deals with investigation of the effect of fabric reinforcement ratio, anchorage length, and surface treatment on concrete composite panels reinforced with carbon fabric under (3 or 4??) point bending. The topic is of interest for journal readers. In general it should be clarified that your anchorage length variable is related to end debonding, while cracks opening along the bent portion of the beam activate local (intermediate) debonding that is sensitive only to the bond characteristics and not to the end anchorage…

This contribution is considered significant since it provides valuable experimental results and the discussion is quite clear and comprehensive along with the theoretical analysis. Howsoever there are still some significant issues that the authors should solve by revising the manuscript before publication.

Some point comment for guiding them are provided in the following:

Please clarify the meaning of symbols x and o in table 1.

I suggest to use “2. Experimental activity” instead of “2. Experiment” section title.

In section 2 please be consistent, use always type or variable (I would suggest “parameter”).

At line 162, “boned” should be “bonded”

According to eq 3 please clarify in the text that tensile strains are negative. Please note that there is an error in equation 5, the fiber contributions should have a minus before ????, since their contribution (below neutral axis) is positive to the bending moment.

At line 200, please revise “In this The deflection of the”

At line 213, please check and add “steel”? In “also STEEL reinforced concrete”…

In section 4, please, if available, add also CoV to the average of the three repetitions for each variable.

Captions of table 5 and Figure 6: is it Four-point or Three-point???

Please note that not perfect bond assumption can be simulated, if not by detailed analysis (e.g. https://doi.org/10.3390/ma13010164 ), by assuming a reduced Young elastic modulus for the fibers.

Please clarify how did you plot figures 11. In the theoretical section you cited a sectional analysis, so it is not obvious how do you move from a cross section of beams to global behaviour.

Author Response

(The authors gave the same response as above.)

Round 2

Reviewer 1 Report

This manuscript is related to the effect of reinforcement ratio and bond characteristic on flexural behavior of carbon textile reinforced concrete panels

Figures 2 and 8 missed the unit

Reviewer 2 Report

Authors adequately solved my concerns.

I feel that "three-point bending test" is better than "third point bending test", however figure 2 shows clearly a "four-point bending test" setup, so I suggest to use only this last term throughout the manuscript
